# Eyewitness accuracy and retrieval effort: Effects of time and repetition

**Philip U. Gustafsson** *, **Torun Lindholm, Fredrik U. Jönsson**

Department of Psychology, Stockholm University, Stockholm, Sweden

\* philipgustafssonresearch@gmail.com

## Abstract

An important task for the law enforcement is to assess the accuracy of eyewitness testimonies. Recent research show that indicators of effortful memory retrieval, such as pausing and hedging (e.g. "I think", "maybe"), are more common in incorrect recall. However, a limitation in these studies is that participants are interviewed shortly after witnessing an event, as opposed to after greater retention intervals. We set out to mitigate this shortcoming by investigating the retrieval effort-accuracy relationship over time. In this study, participants watched a staged crime and were interviewed directly afterwards, and two weeks later. Half the participants also carried out a repetition task during the two-week retention interval. Results showed that the retrieval-effort cues *Delays* and *Hedges* predicted accuracy at both sessions, including after repetition. We also measured confidence, and found that confidence also predicted accuracy over time, although repetition led to increased confidence for incorrect memories. Moreover, retrieval-effort cues partially mediated between accuracy and confidence.

**Data Availability Statement:** The preregistration, materials (in original form), data and code to the analyses in this study are all available at https://osf. io/4hnkp/. Please note however that the staged

## Introduction

The question of when to trust an eyewitness testimony has been a long-standing topic for law-enforcement workers and forensic-psychology researchers alike. One aspect is ascertaining that the eyewitness is not lying. However, a sincere witness may also be incorrect. Thus, it is imperative to find a method not only to detect deception (for meta-analyses see [1–3]), but also to evaluate the accuracy of *sincere* eyewitness statements in testimonies. Recently, Lindholm, et al. [4] examined to what extent metacognitive cues could predict eyewitness accuracy. They found that incorrect statements, as compared to correct statements, were produced with more expressions of effort, such as pauses, fillers (e.g. "uh", "let me see now") and hedges ("maybe", "perhaps"; see also [5–7]). Thus, retrieval-effort cues predicted episodic memory accuracy in an eyewitness context. This study has since been replicated (see [8]). Moreover, research by Gustafsson, et al. [9] indicates that a method relying on effort cues can be used to improve people's judgments of eyewitness accuracy. However, these previous studies have only examined the relationship between retrieval effort and memory accuracy in testimonies given in direct connection to the witnessed event, and it is not known to what extent this relationship holds under other conditions. Retention time and repetition are two major factors

crime video is not included, as not all actors approved global sharing.

**Funding:** Compensation for participation in the study was supported by a grant from the Elisabeth and Herman Rhodin Memorial Foundation (stiftelsemedel.se/stiftelsen-elisabeth-och-herman-rhodins-minne) awarded to PUG. Compensation for the coders was supported by a grant from the Magnus Bergvall Foundation (www.magnbergvallsstiftelse.nu) awarded to TL. There was no additional external funding received for this study. The funders had no role in study design, data collection and analysis, decision to publish, or preparation of the manuscript.

**Competing interests:** The authors have declared that no competing interests exist.

known to affect memory, and that could also be assumed to influence retrieval effort. Specifically, *time* typically leads to forgetting [10], making memories more effortful to retrieve, whereas *repetition* can lead to a better memory and more fluent retrieval [11]. Both factors could thus potentially affect the retrieval effort-accuracy relationship. The major goal of the current study was to examine how time and repetition might influence the relationship between retrieval effort and memory accuracy.

## Retrieval effort and accuracy

There are now many studies that attest to a relationship between retrieval effort and accuracy, in which correct memories tend to be retrieved more fluently than incorrect memories. For example, when participants in two experiments by Brewer, et al. [12] watched a staged crime and had to identify the culprit in a line-up, those who made correct identifications responded quicker than those who made incorrect identifications. Similarly in a recall study, Smith and Clark [7] showed that correct verbal responses to general knowledge questions were quicker, while incorrect responses contained more pauses, fillers and hedges (see also [4–6, 8, 13–16]). Thus, correct memories tend to spring to mind more easily than incorrect memories. Moreover, people seem to have internalized this as a belief. This is evident from studies showing that people judge easily retrieved memories as more likely to be correct, which has been shown both for judgments of one's own memories [17, 18] and for judgments of others' memories [9, 19] (see also [20]).

## Retrieval effort and confidence

Confident, but incorrect eyewitnesses have contributed to many innocent convictions [21, 22]. Thus, more knowledge about factors influencing confidence judgments will enable us to better understand when to trust these judgments.

Given the retrieval effort-accuracy relationship, it is not surprising that research show a relation between retrieval effort and confidence, such that people are more confident in easily retrieved memories [4, 8, 13, 15, 18, 23, 24].

However, research also clearly shows that retrieval ease and confidence judgments do not always correspond with a memory's accuracy. Thus, confidence can increase when retrieval ease is increased, even when accuracy is kept constant. For example, Kelley and Lindsay [17] asked participants to answer general knowledge questions, and manipulated retrieval ease by exposing some participants to potential answers beforehand. Participants were shown answers that were either a) correct, b) incorrect but related, or c) incorrect and unrelated. They measured participants' reaction times, and answers that participants had been exposed to beforehand were provided quicker than non-exposed answers, both for correct and incorrect answers. Moreover, participants gave higher confidence judgments when they provided answers that they had been exposed to beforehand, regardless of whether these answers were correct or incorrect. Thus, exposing people to answers led to a greater retrieval ease, which in turn increased confidence.

Findings by Lindholm et al. [4] corroborate the idea that retrieval effort can act as a variable mediating the relation between accuracy and confidence. In two studies, participants were shown a video of a staged crime, and were then interviewed as eyewitnesses. Each individual statement in these testimonies were then analyzed with regard to signs of retrieval effort and accuracy. The results showed that participants used more expressions suggesting effortful retrieval when memories were incorrect as compared to correct. That is, participants used more *delays*, *fillers* ("uh", "well"), *hedges* ("possibly", "I think") as well as more *words* when describing incorrect details about the witnessed event. The results also showed that participants'

confidence predicted accuracy, but this relation disappeared when effort cues were added to the model. A mediation analysis showed that the effort cue *Hedges* completely mediated the relationship between confidence and accuracy. Although other studies show that confidence contributes uniquely to accuracy in models containing cues to retrieval effort [8, 13, 18], these studies together are in line with the idea that retrieval effort acts as a cue for confidence.

The cue-utilization framework (see [25, 26]) offers a theoretical explanation for the relationship between memory accuracy, retrieval effort and confidence. This framework suggests that people generally lack the ability to get a direct readout of a memory's "strength", and that they therefore rely on cues relating to strength, such as the retrieval effort, to estimate memory strength (see also [27]). Thus, the ease with which a memory comes to mind serves as a cue to that memory's accuracy. Many of these cues are believed to be automatic and therefore shape our metacognitive judgments outside of our conscious control. However, according to the cue-utilization framework, we can also deliberately adjust our metacognitive judgments based on knowledge and beliefs, such as eyewitnesses increasing their confidence that the perpetrator was a man, due to the knowledge that most offenders are men.

If the cue-utilization framework holds true, it follows that confidence (as well as other metacognitive judgments) should only be accurate as long as the cues it is based on have good predictive validity. This merits further investigation into conditions in which retrieval effort can predict accuracy.

## Retrieval effort and retention time

With time, people risk becoming overconfident, resulting in a diminished confidence-accuracy relationship. For example, an initial confidence judgment (e.g. "I'm not very confident") may accurately reflect identification performance (i.e. identified the wrong person), but through the course of time, a witness may become certain ("I'm very confident"), but the accuracy is not likely to change (i.e. still identifies the wrong person). This is not just a hypothetical example, as evidenced from several real-life cases of mistaken eyewitness identifications (see [21]), in addition to lab experiments [28–30]. The increased confidence over time has also been shown in recall studies [23, 24, 31, 32] (cf. [33, 34]), thus showing a similar pattern for both memory recall and memory recognition. Indeed, the confidence-accuracy relationship appear similar across both recall and recognition [18].

Of course, time itself is not a causal factor [for overconfidence], but merely a facilitator of processes that may cause variations in memory and confidence. So which processes *do* cause the overconfidence then? A possible fundamental factor, given the accuracy-effort-confidence relation, is the ease with which a memory is retrieved, and a subsequent increased confidence from increased retrieval ease. This is sometimes referred to as the *memory strength* increasing (see e.g. [35, 36]). With time, memories, correct as well as incorrect, can both be strengthened—resulting in an easier retrieval, and weakened—leading to a more difficult retrieval. Below, we detail some of the potential effects that strengthened and weakened memories can have on retrieval effort.

**Memory strengthening.** Both correct and incorrect memories can be strengthened over time, making them easier to recall. A basic process that strengthens memory is repetition (e.g. [11, 37–39]). One important form of repetition that strengthens memory is repeated *retrieval* of a memory, generally known as the testing effect. This has consistently been shown to lead to better retention and retrieval compared to a mere repeated exposure (e.g. [40]; see [41] for a meta-analysis), although retrieval can also increase belief in false information if presented after the retrieval attempt (e.g. [42]).

The strengthening of a memory by repeated retrieval could also be assumed to lead to an easier retrieval of the memory. This could affect the relationship between retrieval effort and

accuracy in two ways. First, repetition should facilitate retrieval of correct and incorrect memories alike, reducing the difference in effort between correct and incorrect memories. That is, given that correct memories should initially be easily retrieved (e.g. containing only a few or no pauses and hedges in a statement), the gain in retrieval ease by repetition could be assumed to be small overall. Incorrect memories on the other hand, initially recalled with more effort (e.g. containing several pauses and hedges) should become increasingly easier to retrieve with repetitions. This means that incorrect memories should approach the "effort levels" of correct memories, decreasing the usefulness of retrieval effort as a predictor of memory accuracy.

**Memory weakening.**   Over time, memories that aren't strengthened by repetition may weaken, leading to more difficult retrieval. With time, some memories will also pass a "recall threshold" and become forgotten [10]. The memories most likely to be forgotten are those that are difficult to retrieve already at the initial recall. As incorrect memories by default should be more difficult to retrieve than correct memories, a greater number of incorrect memories can be forgotten. This will likely decrease the difference in effort to retrieve correct and incorrect memories, respectively. The reason for this is that even though most correct memories should become more difficult to retrieve over time, most of them will still be retrieved. For incorrect memories, this will not be the case, as some will be forgotten, leading to a smaller *relative* increase in effort overall, compared to correct memories. That is, the pool of incorrect memories at a second recall attempt should mainly contain memories that were initially easy to recall, and fewer of those that were difficult to recall. This would then make retrieval effort a less useful predictor of eyewitness accuracy. We expect confidence to follow a similar trend, such that one becomes less confident over time, but with a greater confidence decrease for correct memories.

**Factors beyond memory strength.**   Although memory strength/ease of retrieval largely plays a large part in memory recall, there are also other factors that influences judgments about memories, and willingness to report them. Key factors can be grouped around effects on the belief in a memory's accuracy. For example, leading questions (e.g. [43]) and post-event feedback [44] (for identification research, see meta-analysis [45]) may bolster one's belief that a memory is correct, and increase the chances that it is reported (and subsequent confidence judgment). Similarly, suggestions that a memory is incorrect would likely decrease the chance of a witness reporting it, without having a direct detrimental effect on retrieval ease.

It is also important to highlight that the increased effort to report memories that have been weakened over time may be offset by changes in grain-size reporting (see e.g. [46–48]). For example, if the memory of a fine-grained detail (e.g. "green sport jacket") becomes weakened over time (leading to a more effortful retrieval), the witness may opt to report a more coarse-grained version of that detail instead (e.g. "jacket"). The expected increased effort to report the fine-grained details of that memory is therefore likely lost, resulting in a more easily retrieved memory. It is therefore possible that memory *quality* may be diminished over time due to a loss of detail, whereas the quantity might remain constant (see [49]), as well as the retrieval effort.

Nonetheless, taken together, time will allow for processes that may both strengthen and weaken memories, which could lead to a greater difficulty in distinguishing the accuracy of statements in eyewitness testimonies. A somewhat limited amount of empirical research on this topic justifies a further examination into how time and repetition may affect eyewitness accuracy, and specifically the relationship between accuracy and memory retrieval effort.

## The present study

In this study, we will expand on previous research regarding the relationship between accuracy and cues to retrieval effort (e.g. [4, 7, 8]) by investigating how this relationship is affected by

time and repetition. Specifically, participants will watch a staged video crime and recall details of the event, and judge confidence in reported details. They will then return two weeks later and report the details of the event again, including confidence. During the retention interval, half the participants will be asked to repeatedly retrieve the event (repetition condition), while the other half will receive no extra instructions (no-repetition condition). Given the arguments concerning the weakening and strengthening of memories above, we hypothesize that there will be: 1) *A main effect of accuracy on cues to retrieval effort*; incorrect memories will overall include more effort cues compared to correct memories; 2) *a main effect of repetition on cues to retrieval effort*; participants who have engaged in repeated memory retrieval (repetition condition) will report memories with less effort cues in an interview two weeks after having viewed an event, compared to participants with no memory repetition (no-repetition condition); 3) *an interaction between repetition and time on retrieval-effort cues;* the repetition group will report memories with less effort cues in an interview two weeks later, compared to directly after an event, whereas the no-repetition group will report memories with more effort cues two weeks later, compared to directly after the event; 4) *an interaction between accuracy and repetition on retrieval-effort cues;* the difference in retrieval effort between correct and incorrect memories reported after two weeks retention will be smaller for the repetition group compared to the no-repetition group; 5) *an interaction between repetition, time and accuracy on retrieval-effort cues;* the difference in effort cues between correct and incorrect memories will be smaller for the repetition group two weeks retention compared to directly after an event. However, there will be no difference in relative effort reported in correct and incorrect memories over time for the no-repetition group. Moreover, we expect 6) *that effort cues will mediate the relation between memory accuracy and confidence*, and 7) *a main effect of repetition on memory quantity;* the repetition group will report more details in an interview two weeks after having viewed an event, compared to the no-repetition group. Finally, we expect hypotheses 1–5 to hold also for the relationship between accuracy and confidence, but inverted (i.e. higher confidence for correct memories).

## Data availability

The preregistration, materials (in original form), data and code to the analyses in this study are all available at https://osf.io/4hnkp/. Please note however that the staged crime video is not included, as not all actors approved global sharing.

## Method

### Participants

Fifty-six Swedish speakers with normal or corrected-to-normal vision took part in this experiment ($M_{age}$ = 29.45, $SD_{age}$ = 8.22, 67.86% women) in exchange for one (no-repetition group) or two (repetition group) movie vouchers. Participants were informed that they would see a film that contained violence, and that they would later be videotaped in an interview about what they had seen. They were also informed that participation would involve two sessions, spaced two weeks apart. All participants gave written informed consent to participate. The experiment was preregistered (https://osf.io/623rt), and has been approved by the Swedish Ethical Review Authority (#2018/2030-31/5).

The sample size was motivated mainly by a desire to have data from a "large" population, as an a-priori power analysis suggested a smaller sample size. In short, we analyzed data on statement-level, and we expected to receive a total of about 7000 statements from 50 participants, whereas a 95% powered study with a medium effect size required only 153 data points (see preregistration for a detailed overview).

## Materials & procedure

In the first session ("T1"), participants arrived at the lab individually and watched a film on a computer screen of a mock crime where two people assaulted another person. All participants were then interviewed directly about the contents of the film. The interview started with an open free recall task ("I would like you to start by freely recounting what you have seen") followed by open direct questions (e.g. "How was the first perpetrator dressed?"). No specific instruction regarding grain size of reporting was given. As the participants responded to the questions, the interviewer wrote down the answers on a numbered sheet. Following the interview, the interviewer read their answers aloud and the participants rated their confidence in the accuracy of each answer, on a scale from 0% to 100% with integers of 20. All interviews were filmed. Participants then filled out a survey about demographic information. Participants were then randomly assigned to either the repetition or the no-repetition condition. A date for the second session was scheduled for both groups. Participants in the repetition condition were informed that they would be asked to write down all their memories of the event at four occasions between the current and the second session two weeks later. This was done by the use of an online survey that was e-mailed to the participants in intervals of 2–3 days. Participants in the no-repetition condition received no such task.

In the second session two weeks later ("T2"; range = 13–15 days, $M$ = 14.00 days, $SD$ = 0.57), the participants returned to the lab. They first filled out a questionnaire with two manipulation-check questions asking them how much they had reflected back on the event during the two-week interval ("*In the last two weeks, how often have you reflected back on the event that you saw?*" and "*In the last two weeks, how much time have you spent reflecting back on the even that you saw?*"). The first question was answered on an ordinal scale (1 = Never, to 7 = Several times each day), and the second question on a Likert scale (0 = No time at all, to 100 = A lot of time). An interview identical to the first session was then conducted, again followed by confidence ratings of the answers. No film was shown this second session. Finally, participants answered three questions regarding what made them feel confident in their memories (this was not part of the current experiment and will therefore not be reported further). This concluded the participation.

The filmed interviews were transcribed verbatim. Accuracy and effort cues in statements were coded from these transcriptions. All responses were coded, both the free recall and the cued recall questions (cf. [4, 8]). First, two coders, blind to the purpose of the study, picked out statements that were objectively possible to confirm or reject. For example, the statement "*he had a green coat*" is an objectively verifiable detail, whereas "*he was handsome*" is not. Responses that contained several details pertaining to the same object/event were coded as one statement when produced in a clustered fashion (e.g. "He wore dark blue jeans"), and coded as independent statements when produced in a dispersed fashion (e.g. "He wore jeans." // "I think they were dark blue"). The coders first coded 10% of the material together (exact overlap = 82.19%), and one coder then coded the remaining 90%. Next, the same two coders coded memory accuracy. Statements were only coded as correct if they contained wordings that correctly matched the description in the coding sheet. For certain details, several accepted terms were provided in the coding sheet (e.g. hair color as "black" and "dark", pants color as "dark", "dark blue", and "black"). Adverbs of degree (e.g. "quite", "somewhat") were to be ignored in the judgment about accuracy, unless specifically stated in the coding sheet. Statements that did not match the wording in the coding sheet (e.g. "light color" for a detail that was "white") were excluded. Statements that contained both correct and incorrect details were also excluded (e.g. "*He had a black beard*", where "black" = correct, and "beard" = incorrect). Coders again coded 10% of the material together (exact overlap = 82.15%, $\kappa$ = .69) and one of them coded the

remaining 90%. This yielded a total of 6238 correct and 2050 incorrect answers. The coders also counted the number unique details of the event, in each testimony, using the same 10% - 90% setup (exact overlap = 83.15%, $\kappa$ = .61). For example, the detail "black beard" was counted as one unique detail, even if a witness mentioned it in several statements. Two new coders then coded effort cues in the statements. They coded 10% of the material together and one of the coders coded the remaining 90%. The coding followed the proceedings by Gustafsson et al. [8]: 1) *Delays*–a pause longer than two seconds before or during a response (exact overlap = 100%, $\kappa$ = 1.0); 2) *Non-word fillers*–interjections and sounds like "hm", "uh" (exact overlap = 97.58%, $\kappa$ = .99); 3) *Word fillers*–"meaningless" words like "you know", "well" (exact overlap = 73.21%, $\kappa$ = .97). This category also includes self-talk "Let's see. . ."; 4) *Hedges*–word forms that reduce the force of an assertion, allow for exceptions, or avoid commitment, such as "I think", "maybe" (exact overlap = 87.50%, $\kappa$ = .98). Additionally, two new effort cues were coded, inspired from psycholinguistic research [50, 51]: 5) *False starts*–initiated expressions that are stopped and then started anew, such as "He wher-, he wore a hat" (exact overlap = 76.65%, $\kappa$ = .97); and finally, 6) *Prolongations*–prolonged pronunciations of a word (exact overlap = 99.76%, $\kappa$ > .99).

## Data analysis

For all analyses of retrieval-effort cues and accuracy, we decided a priori to group together free recall and cued recall responses (see preregistration). This was done to limit the complexity of the research design, which already involved three-way interactions on multiple outcome variables (see Hypotheses 3–5). For the interested reader, posteriori plotting of free and cued recall results separately are found in the supplementary section (S1 and S2 Figs). For further convenience, these analyses were also only carried out on statements for which we had also obtained confidence judgments ($N$ = 5918; $n_{correct}$ = 4397, $n_{incorrect}$ = 1521). Alpha-level was set to 0.05 for all analyses. Finally, all analyses were carried out with Rstudio [52] in R [53]. Multilevel analyses were carried out with the *lme4* [54] and *lmertest* packages [55]; the mediation analysis was done with the *mediation* package [56]; and the figures were made using the *tidyverse* package [57].

## Results

### The repetition group reflected more on the witnessed event

We first checked whether repeatedly retrieving memories of the event (i.e. the repetition condition) led participants to think more about the event compared to the no-repetition group. Two between-groups t-tests with repetition (repetition / no repetition) as independent variable and the respective manipulation-check question as dependent variable supported this notion. The repetition group ($M$ = 3.07, $SD$ = 0.55) had thought *more often* about the event relative to the no-repetition group ($M$ = 2.21, $SD$ = 0.56), $t(53.84)$ = 5.85, $p$ < .001, $d$ = 1.56, and also spent *more time* thinking about the event, $t(53.98)$ = 2.93, $p$ = .005, $d$ = 0.82 ($M_{repetition}$ = 27.67, $SD$ = 12.23; $M_{control}$ = 17.62, $SD$ = 13.41).

### Effects of accuracy, time and repetition on amount of details

In our main analyses, we first examined the effects of accuracy, time and repetition and interactions on amount of unique details mentioned by the witnesses. This was done separately for free and cued recall, as well as for the total amount of unique details (i.e. free and cued recall combined, ignoring overlapping details). While examinations of free and cued recall were done exploratively, we hypothesized a greater number of total recalled details for the repetition group at the second interview two weeks later, compared to the no-repetition group (Hypothesis 7).

An ANOVA with *Amount of unique details during free recall* showed statistically significant effects of all dependent variables: *Repetition*, $F(1, 54) = 31.29$, $p < .001$, $\eta^2 = .054$ ($M_{repetition} = 9.84$, $SD = 9.87$; $M_{no-repetition} = 5.97$, $SD = 6.03$); *Time*, $F(1, 54) = 49.02$, $p < .001$, $\eta^2 = .085$ ($M_{T1} = 5.42$, $SD = 5.92$; $M_{T2} = 10.26$, $SD = 9.60$); and *Accuracy*, $F(1, 54) = 234.72$, $p < .001$, $\eta^2 = .407$ ($M_{correct} = 13.13$, $SD = 8.19$; $M_{incorrect} = 2.54$, $SD = 3.93$), see Fig 1. All interactions also showed statistically significant effects: *Repetition-Time*, $F(1, 52) = 28.27$, $p < .001$, $\eta^2 = .049$; *Repetition-Accuracy*, $F(1, 52) = 6.06$, $p = .015$, $\eta^2 = .010$; *Time-Accuracy*, $F(1, 52) = 7.22$, $p = .008$, $\eta^2 = .013$; and *Repetition-Time-Accuracy*, $F(1, 50) = 4.57$, $p = .033$, $\eta^2 = .008$), see Fig 1. Post-hoc analyses with Bonferroni corrections showed no statistically significant difference in amount of details for the repetition group and the no-repetition group at the first session (T1; $M_{diff} = 0.19$, $p = .866$, $d = 0.03$). The repetition group reported a significantly greater number of details two weeks later (T2; $M_{diff} = 8.65$, $p < .001$, $d = 1.35$), whereas the increase was not statistically significant for the no-repetition group ($M_{diff} = 1.29$, $p = .250$, $d = 0.23$). The increased amount of details for the repetition group at T2 was mainly driven by an increase in incorrect responses ($M_{diff} = 5.26$, $p < .001$, $d = 4.53$), which was greater than the also significant increase of correct responses ($M_{diff} = 12.04$, $p < .001$, $d = 1.90$). For the no-repetition group, both amount of correct ($M_{diff} = 1.72$, $p = .233$, $d = 0.36$) and incorrect details increased, but the results were not statistically significant ($M_{diff} = 0.86$, $p = .042$, $d = 0.74$).

An ANOVA with *Amount of unique details during cued recall* showed statistically significant effects of *Time*, $F(1, 54) = 4.48$, $p = .035$, $\eta^2 = .005$ ($M_{T1} = 23.58$, $SD = 12.12$; $M_{T2} = 21.91$, $SD = 10.38$); and *Accuracy*, $F(1, 54) = 640.40$, $p < .001$, $\eta^2 = .735$ ($M_{correct} = 32.38$, $SD = 7.14$; $M_{incorrect} = 13.06$, $SD = 4.09$); but not *Repetition*, $F(1, 54) = 0.05$, $p = .822$, $\eta^2 < .001$ ($M_{repetition} = 22.63$, $SD = 10.94$; $M_{no-repetition} = 22.80$, $SD = 11.65$), see Fig 1. There was also a statistically significant interaction between *Time* and *Accuracy*, $F(1, 52) = 7.56$, $p = .006$, $\eta^2 = .009$, but not *Repetition-Time*, $F(1, 52) = 0.91$, $p = .341$, $\eta^2 = .001$; *Repetition-Accuracy*, $F(1, 52) = 0.75$, $p = .389$, $\eta^2 = .001$; nor *Repetition-Time-Accuracy*, $F(1, 50) = 0.06$, $p = .807$, $\eta^2 = .001$. A post-hoc analysis with Bonferroni correction showed a statistically significant decrease of correct details at T2 compared to T1 ($M_{diff} = 3.71$, $d = .53$, $p = .005$, see Fig 1). Incorrect details instead increased over time, but the effect was not statistically significant ($M_{diff} = 0.48$, $d = 0.13$ $p = .536$, see Fig 1).

Finally, an ANOVA with *Total amount of unique details* showed a statistically significant effect of *Accuracy*, $F(1, 54) = 753.16$, $p < .001$, $\eta^2 = .771$ ($M_{correct} = 37.00$, $SD = 7.12$; $M_{incorrect} = 14.38$, $SD = 5.09$) but no statistically significant effect of *Time*, $F(1, 54) = 0.54$, $p = .542$, $\eta^2 = .001$ ($M_{T1} = 25.38$, $SD = 13.25$; $M_{T2} = 25.99$, $SD = 12.61$) nor *Repetition*, $F(1, 54) = 3.37$, $p = .068$, $\eta^2 = .003$ ($M_{repetition} = 26.47$, $SD = 12.94$; $M_{no-repetition} = 24.96$, $SD = 12.89$). None of the interactions were statistically significant, *Repetition-Time*, $F(1, 52) = 1.19$, $p = .278$, $\eta^2 = .049$;

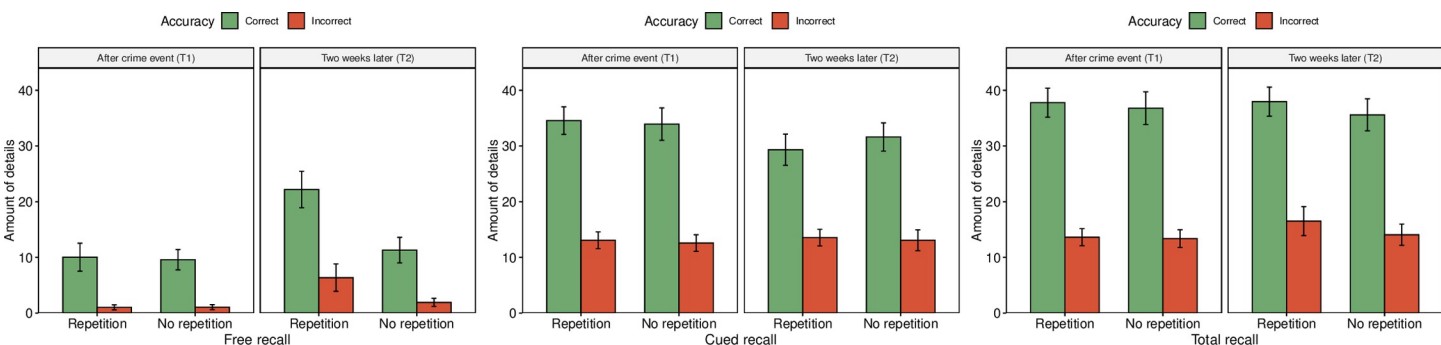

**Fig 1. Mean number of unique correct and incorrect details recalled across time and repetition.** Note. Error bars represent 95% confidence intervals.

*Repetition-Accuracy*, $F(1, 52) = 0.04$, $p = .841$, $\eta^2 < .001$; *Time-Accuracy*, $F(1, 52) = 1.92$, $p = .167$, $\eta^2 = .002$; *Repetition-Time-Accuracy*, $F(1, 50) = 0.06$, $p = .807$, $\eta^2 < .001$, see Fig 1. To test our hypothesis that repetition led to retrieval of more details (Hypothesis 7), we carried out a planned comparison between the repetition group and the no-repetition group at the second session ("T2"). Results showed no statistically significant effect of *Repetition* on *Total amount of unique details* at T2, $M_{diff} = 4.82$, $p = .052$, $d = 0.52$ (see Fig 1).

## Retrieval-effort cues and confidence predict memory accuracy

Next, we examined whether the effort cues (and confidence) predicted accuracy. We used multilevel modelling with statements nested within witnesses (i.e. participant as random factor; [58]), and compared a baseline, intercept-only model of accuracy, with models containing each effort cue (and confidence) as predictors. We expected fewer effort cues and greater confidence for correct memories (Hypothesis 1). If this hypothesis were to be supported in the data, we would expect a) that predictor models would differ significantly from baseline-models (i.e. p < .05), and b) have a greater Akaike weights ($w_{i}(AIC)$) than the baseline models (i.e. > .5; for more on Akaike weights, see [59, 60]). In line with expectations, comparisons between the baseline and predictor models showed that model fit was significantly improved when adding the effort cues *Delays*, *Hedges*, *Non-word fillers*, and *Word fillers* and *Confidence* (see Table 1 and Figs 2 and 3). However, there were no statistically significant differences between baseline models and the predictor models with *False starts* and *Prolongations*, respectively (see Table 1 and Fig 2). The results for delays, hedges and word fillers thus supported Hypothesis 1, that is, a greater amount of effort cues in incorrect rather than correct statements. Likewise, as expected, confidence was greater for correct rather than incorrect statements. However, contrary to expectations, false starts and prolongations did not significantly predict accuracy. Moreover, non-word fillers were more common in correct rather than incorrect responses, thus predicting accuracy opposite to the expected direction.

Following the procedures of Lindholm et al. [4], we then created a single model containing all the significant predictors and examined their relative contribution to accuracy. Thus, we created a model with *Delays*, *Hedges*, *Non-word fillers*, *Word fillers and Confidence*. No explicit prediction was made for this analysis. All predictors except *Word fillers* proved significant, unique predictors of accuracy in the resulting model (see Table 2). In Table 2, the odds ratio indicates the increase/decrease in accuracy when increasing one step on the scale of each variable, with values above zero indicating an increase, and values below zero indicating a decrease. That is, a statement with no hedges will be 29% more likely to be correct compared to a statement with one hedge (UOR = 0.71), and a confidence judgment with 81% confidence is 3% more likely to be correct compared to a confidence judgment of 80% (UOR = 1.03; see Table 2).

**Table 1. Results of model comparisons assessing the effects of each retrieval-effort cue and confidence on memory accuracy.**

| *Effort Cues* | Test statistics | | |
|---|---|---|---|
| Delays | $\chi^2(1, N = 5918) = 45.50$ | $p < .001$ | $w_{i}(AIC) > .99$ |
| Hedges | $\chi^2(1, N = 5918) = 195.57$ | $p < .001$ | $w_{i}(AIC) > .99$ |
| Non-word fillers | $\chi^2(1, N = 5918) = 5.68$ | $p = .017$ | $w_{i}(AIC) = .86$ |
| Word fillers | $\chi^2(1, N = 5918) = 10.00$ | $p = .002$ | $w_{i}(AIC) = .98$ |
| False starts | $\chi^2(1, N = 5918) = 0.05$ | $p = .830$ | $w_{i}(AIC) = .28$ |
| Prolongations | $\chi^2(1, N = 5918) < 0.01$ | $p = .962$ | $w_{i}(AIC) = .27$ |
| *Confidence* | $\chi^2(1, N = 5918) = 471.06$ | $p < .001$ | $w_{i}(AIC) > .99$ |

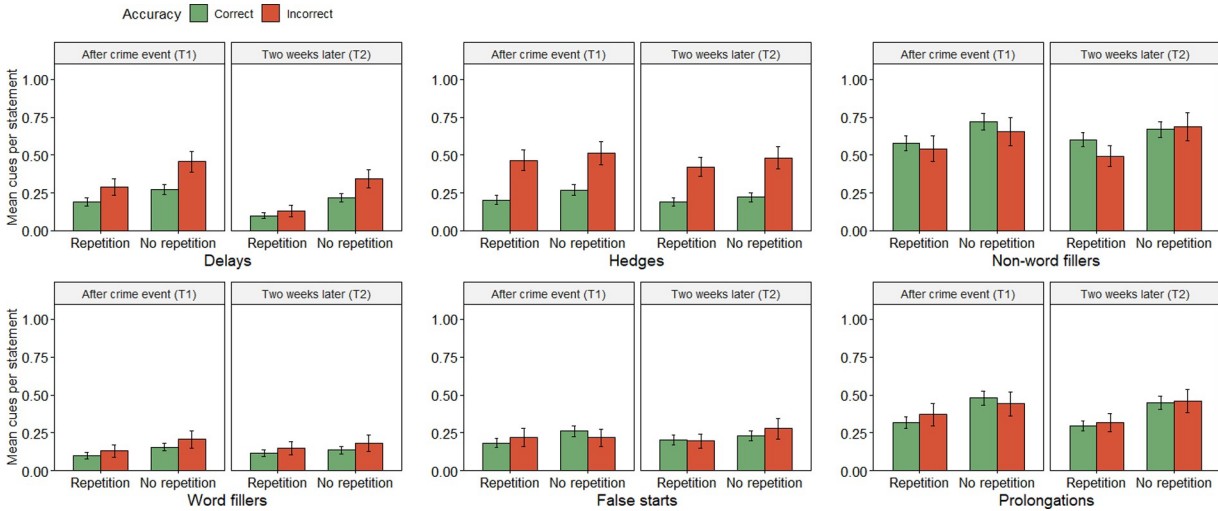

**Fig 2. Effects of accuracy, time and repetition on retrieval-effort cues.** Note. Error bars represent 95% confidence intervals.

We next examined the effects of time and repetition on the relationship between accuracy and retrieval-effort cues, as well as confidence (Hypotheses 2–5). To make these analyses less convoluted, we first created an "effort index" out of the effort cues that had significant unique contribution to accuracy (see preregistration). This included *Delays*, *Hedges* and *Non-word fillers*. However, as the effect of non-word fillers on accuracy was opposite to the expected direction (see Fig 2), and contrary to previous findings [4, 7, 8], we decided to drop non-word fillers and make the effort index out of the two remaining cues: *Delays* and *Hedges*.

## Effects of time and repetition on retrieval-effort cues

To test the effects of time and repetition on retrieval-effort cues, we compared a baseline, intercept-only model of the effort index with models containing *Time*, *Repetition*, *Accuracy* and their interactions as predictors. For these analyses we expected a main effect of accuracy (a greater effort index value for incorrect memories, Hypothesis 1), and then several interactions (Hypotheses 2–5). In short, these interactions could be summarized as an easier retrieval over time for the repetition group and a more difficult retrieval over time for the no-repetition group (Hypothesis 2–3), and a smaller difference in retrieval-effort cues between correct and incorrect memories over time for the repetition group (Hypothesis 4–5). As in the previous multilevel analyses, we would expect these differences to manifest in the data as significant predictor models with higher Akaike weights compared to baseline models. Turning to the results then, analyses showed that that model fit was indeed significantly improved compared to the baseline model when adding *Accuracy*, such that incorrect statements ($M = 0.77$, $SD = 0.97$) were produced with more effort compared to correct statements ($M = 0.42$, $SD = 0.75$; $d = 0.44$; see Table 3 and Fig 4). Unexpectedly, model fit was also significantly improved when adding *Time*, such that statements were produced with more effort at the first interview (T1; $M = 0.57$, $SD = 0.88$) compared to the second interview two weeks later (T2; $M = 0.45$, $SD = 0.77$, $d = 0.14$; see Table 3 and Fig 4). Moreover, model fit was significantly improved with *Repetition*, such that the repetition group reported memories with less effort ($M = 0.42$, $SD = 0.73$) compared to the no-repetition group ($M = 0.59$, $SD = 0.90$, $d = 0.23$, see Table 3 and Fig 4). To test interaction effects, we created models containing each interaction and compared them to models of only their respective predictors (e.g. a model with time and accuracy

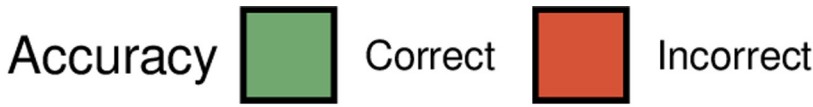

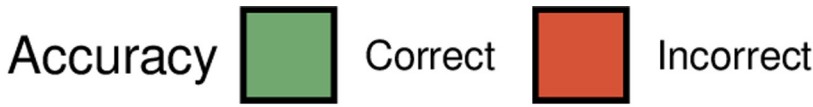

**Fig 3. Effects of accuracy, time and repetition on confidence.** Note. Error bars represent 95% confidence intervals.

as predictors was compared to a model with time, accuracy and the time-accuracy interaction as predictors). Here we expected significant interactions of all four combinations of *Accuracy*, *Time* and *Repetition*. Contrary to expectations however, none of the interactions significantly improved fit (see Table 3).

### Effects of time and repetition on confidence

We then carried out identical analyses for a confidence model. Expectations were identical to those for retrieval effort but with reversed directions, such that we expected *lower* confidence in conditions where we had expected *higher* retrieval effort. In line with predictions, results showed that model fit was significantly improved compared to the baseline model when adding *Accuracy*, such that confidence was higher for correct statements ($M = 86.25$, $SD = 21.00$) compared to incorrect statements ($M = 71.65$, $SD = 26.18$, $d = 0.70$, see Table 3 and Fig 3). Moreover also as expected, model fit was not significantly improved when adding *Time*, ($M_{T1} = 82.45$, $SD = 23.54$; $M_{T2} = 82.54$, $SD = 23.13$ $d < 0.01$, see Table 3 and Fig 3); nor *Repetition*

**Table 2. Multilevel logistic regression analysis predicting memory accuracy from retrieval-effort cues and confidence (z-transformed).**

| | | | | | 95% CI for OR | |
|---|---|---|---|---|---|---|
| Predictor | B (SE) | z | UOR | OR | LL | UL |
| **Delays** | **-0.09 (0.03)** | **-2.68**\*\* | **0.84** | **0.92** | **0.86** | **0.98** |
| **Hedges** | **-0.20 (0.03)** | **-6.19**\*\*\* | **0.71** | **0.82** | **0.77** | **0.87** |
| **Non-word fillers** | **0.08 (0.04)** | **2.40**\* | **1.10** | **1.08** | **1.02** | **1.17** |
| Word fillers | 0.04 (0.03) | 1.24 | 1.08 | 1.04 | 0.98 | 1.11 |
| **Confidence** | **0.59 (0.04)** | **16.61**\*\*\* | **1.03** | **1.80** | **1.68** | **1.93** |
| Model fit[1] | $\chi^2(1, N = 5918) = 522.05$\*\*\*, $w_i(AIC) > .99$ | | | | | |

Note. Parameters whose CI of OR do not include 1 are boldfaced. B = logistic coefficients; SE = Standard errors of the logistic coefficient estimates; z = z-value of coefficient; UOR = unstandardized odds ratio; OR = standardized odds ratio; CI = confidence interval; LL = lower limit of OR; UL = upper limit of OR.

\* $p < .05$,

\*\* $p < .01$.

\*\*\* $p < .001$. [1]Model fit compared to a baseline, intercept only model.

($M_{repetition}$ = 83.53, $SD$ = 21.73; $M_{no\text{-}repetition}$ = 81.47, $SD$ = 24.78, $d$ = 0.09; see Table 3 and Fig 3). Interestingly, all interactions improved fit as expected: *Time-Repetition*, *Time-Accuracy*, *Repetition-Accuracy*, and also *Time-Repetition-Accuracy* (see Table 3 and Fig 3). Planned comparisons (see preregistration) showed that confidence indeed significantly increased from T1 to T2 for the repetition group ($M_{diff}$ = 1.62, $p$ = .042, $d$ = 0.07) whereas there was a nonsignificant decrease for the no-repetition group ($M_{diff}$ = -1.56, $p$ = .087, $d$ = 0.06). Moreover as expected, the increased confidence for the repetition group was mainly driven by higher confidence in incorrect statements ($M_{diff}$ = 6.51, $p < .001$, $d$ = 0.27) as there was no significant increase in correct statements ($M_{diff}$ = 0.53, $p$ = .536, $d$ = 0.03). For the no-repetition group, the decrease in confidence between T1 and T2 was not statistically significant for neither incorrect statements ($M_{diff}$ = -0.61, $p$ = .764, $d$ = 0.02), nor correct statements ($M_{diff}$ = -1.58, $p$ = .091, $d$ = 0.07). Hence, in contrast to the results for the effort index, the results for confidence were more in line with expectations.

**Table 3. Results of model comparisons assessing the effects of time, repetition and accuracy on the retrieval-effort index and confidence.**

| | | Test statistics |
|---|---|---|
| Time | *Retrieval-effort index* | $\chi^2(1, N = 5918) = 24.67, p < .001, w_i(AIC) > .99$ |
| | *Confidence* | $\chi^2(1, N = 5918) = 0.25, p = .620, w_i(AIC) = .27$ |
| Repetition | *Retrieval-effort index* | $\chi^2(1, N = 5918) = 9.63, p = .002, w_i(AIC) = .97$ |
| | *Confidence* | $\chi^2(1, N = 5918) = 1.35, p = .245, w_i(AIC) = .38$ |
| Accuracy | *Retrieval-effort index* | $\chi^2(1, N = 5918) = 211.90, p < .001, w_i(AIC) > .99$ |
| | *Confidence* | $\chi^2(1, N = 5918) = 539.15, p < .001, w_i(AIC) > .99$ |
| Time* Repetition | *Retrieval-effort index* | $\chi^2(1, N = 5918) = 0.17, p = .683, w_i(AIC) = .27$ |
| | *Confidence* | $\chi^2(1, N = 5918) = 6.03, p = .014, w_i(AIC) = .88$ |
| Time* Accuracy | *Retrieval-effort index* | $\chi^2(1, N = 5918) = 2.83, p = .093, w_i(AIC) = .50$ |
| | *Confidence* | $\chi^2(1, N = 5918) = 8.56, p = .003, w_i(AIC) = .95$ |
| Repetition*Accuracy | *Retrieval-effort index* | $\chi^2(1, N = 5918) = 2.83, p = .093, w_i(AIC) = .50$ |
| | *Confidence* | $\chi^2(1, N = 5918) = 5.92, p = .015, w_i(AIC) = .88$ |
| Time* Repetition*Accuracy | *Retrieval-effort index* | $\chi^2(4, N = 5918) = 6.05, p = .195, w_i(AIC) = .38$ |
| | *Confidence* | $\chi^2(4, N = 5918) = 26.23, p < .001, w_i(AIC) > .99$ |

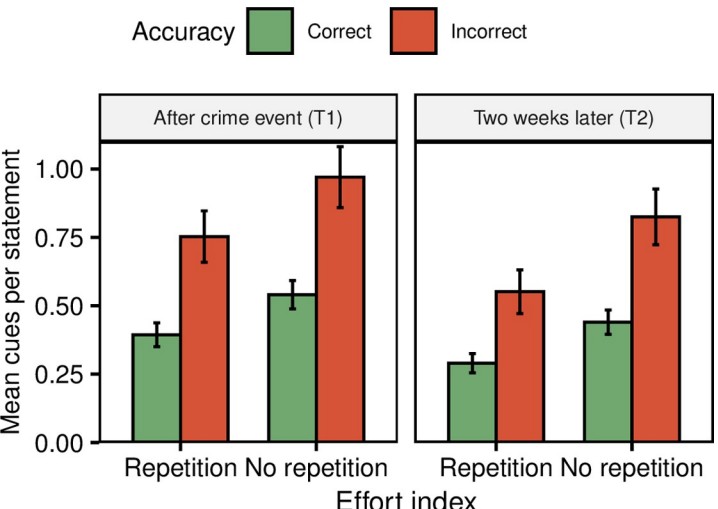

**Fig 4. Effects of accuracy, time and repetition on the retrieval-effort index.** Note. Error bars represent 95% confidence intervals.

## Retrieval-effort index mediates between confidence and accuracy

Finally, to examine whether retrieval effort was used as a basis for confidence (Hypothesis 6), we carried out a mediation analysis between accuracy and confidence, with the effort index as mediator. Results showed that the effort index mediated 22.7% of the relation between accuracy and confidence (see Fig 5).

## Discussion

The main aim of this experiment was to investigate the effects of time and repetition on the relation between retrieval effort and accuracy. Secondary aims involved investigations of confidence, as well as the relationship between retrieval effort, confidence, and accuracy. Additionally, we examined the effects of time and repetition on memory accuracy and memory

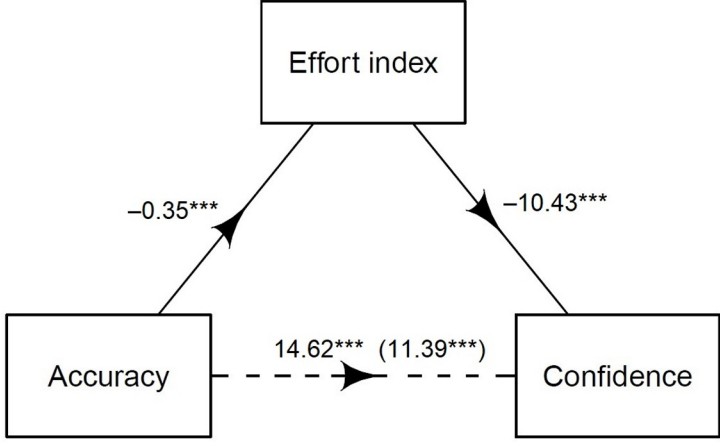

**Fig 5. Mediation analysis between memory accuracy and confidence, with the retrieval-effort index as mediator.**

quantity. There are four major take-aways from this study, namely that 1) retrieval-effort cues predict accuracy over time, 2) retrieval-effort cues decrease over time, 3) confidence increases over time mainly for incorrect memories when memories are repeated, and 4) a retrieval-effort index mediates the relation between confidence and accuracy. However, there were also methodological constraints, leading to smaller than expected effects of time and repetition, that also potentially limits generalizability. We will now detail discussions of each of these findings, before moving on to general discussions and limitations.

## Retrieval-effort cues predict accuracy over time

Overall, our results add to a large body of research showing that correct memories are more easily retrieved compared to incorrect memories (e.g. [4–6, 8, 13–16]; see Figs 2 and 4). Whereas other studies have examined this relationship after retention intervals up to a few minutes, we demonstrate that this relationship persists at a second recall a couple of weeks later. This was evident also when participants had engaged in repeated retrieval during the retention interval (see Fig 4). The results thus suggest that retrieval-effort cues can be reliable predictors of memory accuracy over extended retention intervals (however, see also section "Retrieval ease decreases over time" below).

We measured six cues to retrieval effort, of which *Delays*, *Hedges* and *Word fillers* were significantly more numerous in incorrect responses compared to correct responses. *Delays* and *Hedges* were the strongest predictors (see Table 1), in line with previous findings [4, 8]. Somewhat surprisingly, *Non-word fillers* showed the opposite result, as they were more numerous in correct responses (see Fig 2). Previous findings on non-word fillers have been slightly inconclusive, as Lindholm et al. [4] and Smith and Clark [7] found non-word fillers to be significantly more common in incorrect responses, whereas there was no statistically significant effect in Gustafsson et al. [8]. Although our reasoning has been that non-word fillers are expressed automatically as a consequence of effortful memory retrieval, Clark and Tree [50] compellingly argues that fillers are used intentionally like conventional words in a language, and that their usage largely signal turn-taking. Thus, non-word fillers might not always signal that one is effortfully attempting to retrieve a memory, but may also signal that one is deciding how to formulate a coming sentence, or that one wants to end a speaking turn. This offers an explanation to the inconsistencies in the findings in regards to accuracy, although it is still somewhat puzzling that we found significantly *more* non-word fillers in correct statements. Nonetheless, an important conclusion from these contrasting results is that non-word fillers is not a reliable predictor of accuracy. Another surprising finding was that the two "new" measures of retrieval effort–*Prolongations* and *False starts*–did not significantly predict accuracy (see Fig 2). These two cues were inspired from psycholinguistic research on disfluencies, that is, utterances that disrupt the flow of speech (e.g. [51]). We reasoned that a prolonged pronunciation of a word would be a consequence of an inability to retrieve a memory and therefore be more common in incorrect responses. Similarly, we believed that false starts would mainly occur when a memory was not fully retrieved, and hence signal inaccuracy. We found no support for these ideas however. Instead, the overarching evidence points to *Hedges* and *Delays* as the most reliable effort cues to indicate memory accuracy.

Given the consistent effort-accuracy relation, a reasonable question to ask is how to use this knowledge in the field as a practitioner. Because there appears to be some variation between individuals in expressing effort (see e.g. "T1" in Fig 4) we suggest that this knowledge is as of now best used carefully, ideally in conjunction with other corroborating evidence, such as physical evidence or other eyewitness reports. A starting point could be to judge statements without hedges or delays as correct, which is supported by the unstandardized odds ratios in

Table 2, which show that each hedge should decrease the likelihood of accurate recall by 29%, and each delay should decrease accuracy by about 16%. Such a method has actually shown some success in improving judgment accuracy of eyewitness testimonies (see [9]), and is similar to the recommendation given by Wixted and Wells [61] for identification research, where they suggest that highly confident witnesses should generally be believed (given "pristine" conditions).

## Retrieval-effort cues decrease over time

Another major finding is that retrieval-effort cues decreased over time regardless of repetition and accuracy. That is, participants that had repeatedly retrieved memories over the two-week interval, as well as those that hadn't, used fewer effort cues at the second interview when recalling correct and incorrect statements alike ($d$ = 0.14). We only expected increased retrieval ease for the repetition group, as repetition is known to facilitate retrieval (e.g. [11, 37]), and expected the no-repetition group to have *greater* difficulties to retrieve memories, due to memory weakening and forgetting. The increased retrieval ease for the no-repetition group at T2 is not likely due to spontaneous repetition among these participants, as they scored low on the two questions about time spent reflecting on the event, nor due to selective reporting of easily retrieved memories (see [62]), as there was no significant reduction in the number of total unique details reported between T1 and T2 (see Fig 1). Instead we see three plausible explanations for this effect: a) repeated retrieval opportunities as T1, b) context-dependent learning effects, and c) switch in grain-size of reported details.

The first explanation—repeated retrieval opportunities at T1—is probably the most important. That is, the participants were allowed to retrieve their memories of the event directly after seeing it, as they were interviewed about the event. They did this several times, first in the free recall session and again in the cued recall session. In addition, there was a third retrieval opportunity, namely during the confidence ratings. During this task, the experimenter read aloud details that the witness had reported, and thus allowed the participant to elaborate upon each mentioned detail. This abundance of retrieval attempts likely helped participants consolidate their memory of the event, leading to only minor forgetting at T2 for both groups. Although this could likely have been avoided with a between-group design in which one group were only tested at T2, a within-group was the optimal choice to follow the development of effort cues over time, given the variance in use of effort cues between individuals (see [9], but see also T1 in Fig 4 above).

The second explanation—context-dependent learning [63]—suggests a more successful retrieval when the retrieval takes place at the same location as the encoding of the event. Our participants were interviewed in the same experimental room during both sessions, so it is possible that this context facilitated their retrieval, minimizing potential forgetting effects during the retention interval.

The third explanation is a shift in grain-size of the details. Koriat and Goldsmith [62] suggest that people can not only decide to withhold or report a memory, but also shift the level of detail with which the memory is reported. Thus, a piece of clothing could both be described with a high level of detail ("a blue zip up jacket with green stripes and a hood") or a low level of detail ("a jacket"). In this study we did not code for grain-size, so it is possible that the repetition group remembered things in more detail than did the no-repetition group. We encourage researchers to examine this in future studies, and as our data is openly available, suggest examinations also of our data.

We suggest that these explanations also likely led to a smaller overall effect of the repetition manipulation, due to floor effects.

In addition to expecting an easier retrieval overall at T2 for the repetition group, we also expected an increased retrieval ease for incorrect statements compared to correct statements, again due to expected floor effects for correct memories. The findings indeed indicate a greater retrieval ease for incorrect memories (see Fig 4), but the time-accuracy interaction was not statistically significant. If this effect is not due to changes in grain-size of reporting, one might dismiss it as a too weak manipulation of repetition. However, we believe that this showcases something more important, namely that all memories are not as fragile and easily manipulated as the memory research field might sometimes give the impression of (see also [33, 34]). Finally, although it is worth pointing out again that the effort cues still predicted accuracy at T2 despite the general increase in retrieval ease (see Fig 4), a continued decrease over time will likely lead to a point wherein accuracy can no longer be distinguished by retrieval-effort cues. Thus, examining effects of longer retention intervals with repetition is an avenue for future research.

## Higher confidence in incorrect memories after repetition

Regarding confidence, we found no general increase or decrease over time. Instead we found a three-way interaction between accuracy, time, and repetition, with the notable finding that confidence only significantly increased for *incorrect memories* in the repetition group ($d = 0.27$; see Fig 3). The effect was not evident for correct memories ($d = 0.03$). To the best of our knowledge, this is the first study to demonstrate this effect. Previous studies examining the effect of confidence over time have generally presented an overall confidence score, rather than separate values for correct and incorrect responses (e.g. [23, 24, 30–34]). This is an interesting result, because it would suggest that we should effectively trust people's confidence for correct memories, as these judgments were relatively stable over time and across repetition in our study. From a practical standpoint though, there is of course the big caveat that one generally has access to the confidence but not the accuracy, and that the use of confidence is to *derive* accuracy, not the other way around. Nonetheless, these results could have potential implications for the legal system. For example, Wixted and Wells [61] (see also [64]) have demonstrated that initially confident witnesses should generally be trusted (given "pristine" line-up conditions), as they are often correct, but not initially unconfident witnesses, as they are more likely to be incorrect. Our results add to this research by suggesting that initially confident witnesses could potentially be trusted over time, such as in later interviews with police and jurors, as they should generally remember correctly, and that they are likely to retain similar confidence levels over time. Initially unconfident witness on the other hand would instead be more likely to increase their confidence over time and become overconfident. Some reservations to these suggestions are necessary however, as we did not observe the typical forgetting effect over time, which could indicate that these results are not representative of general situations (see also "Limitations").

The increase in confidence for incorrect but not correct memories after repetition is presumably due to ceiling effects. That is, confidence in correct memories already approached the max rating of 100% during the first session ($M_{correct} = 86.36$, of which 60.71% of those ratings were "100"), and thus had less room for increases than confidence for incorrect memories ($M_{incorrect} = 71.05$, of which 25.59% of those ratings were "100", see S1 Table).

For the no-repetition group, confidence decreased slightly rather than increased over time, but this decrease was not statistically significant for neither correct ($d = 0.07$) nor incorrect memories ($d = 0.02$; see Fig 3). We had expected confidence to decrease over time as a consequence of more difficult memory retrieval due to memory weakening and forgetting (cf. [65]), but as no major forgetting took place (see Fig 3), it is unsurprising that confidence remained relatively stable.

A final take-away from the confidence results is that confidence still predicted accuracy at the second interview two weeks after witnessing the original event, despite the increased confidence in incorrect memories for the repetition group. Thus, similar to the results of the retrieval-effort cues, the change induced by a two-week retention interval (and memory repetition) did not largely disrupt the possibility to predict accuracy from these two variables. This suggests that confidence can remain a reliable predictor over time. It is however important to note that this study was limited to a retention interval of only two weeks, and it is plausible that greater retention intervals and repetitions would eventually lead to an elimination of the confidence-accuracy (as well as the retrieval effort-accuracy) relationship.

## Retrieval-effort index mediates between confidence and accuracy

The final major finding was that retrieval effort mediated between accuracy and confidence. Specifically, an index out of the two effort cues *Hedges* and *Delays*, mediated 22.7% of the relation between accuracy in reported memories and confidence in those memories. We draw two distinct conclusions from these results. First, the results support the cue-utilization view [25, 26], that is, that people make metacognitive judgments such as confidence based on *cues*–in this case cues to retrieval effort, as participants were more confident in memories that were easily retrieved (which in turn were more likely to be correct). Second, the results suggest further bases for confidence in addition to the retrieval cues *Hedges* and *Delays*, given the relatively low percentage mediated (cf. [4, 8]). We have previously argued [8] that leftover variance could potentially be explained by confidence being "information-based", that is, based on knowledge and beliefs (for example, relying on the knowledge that it is hard to see colors accurately at night, when assessing confidence in the memory of a perpetrator's clothing), in addition to the more automatic experience-based judgments from retrieval effort. However, we deem it unlikely that the majority of confidence judgments would be based on knowledge and beliefs, as these judgments are supposedly deliberate (see [26]), which contrasts with the commonly accepted view within metacognitive research that our ability to assess the bases for our metacognitive judgments is highly limited [27]. Instead, a potential explanation is that participants based their confidence on other automatic cues, and perhaps that our measures of retrieval effort did not fully capture the phenomenological experience of finding a memory hard to retrieve (supplemental analyses revealed that the other effort cues in our study were not major mediators either, see S2 Table). A final explanation is that the coding sheet contained unclarities or inconsistencies regarding the effort cues, which could explain the discrepancies between this study and previous studies regarding the effect size of the mediated relationship between confidence and accuracy (cf. [4, 8]). However, the high intercoder agreement seems to suggest the opposite. Nonetheless, to conclude, we did find that an index for retrieval effort partially mediated the relation between confidence and accuracy, corroborating previous research [4, 8]. It remains an endeavor for future research to examine further bases for confidence.

## Memory accuracy and amount of unique details

We also examined the effects of time and repetition on memory accuracy, and on the amount of unique details provided by the witnesses. The biggest finding was an increase of both correct and incorrect details for the repetition group at T2 during free recall (see Fig 1). This suggests that the repetition manipulation did facilitate memory recall at T2, supporting the idea that the repetition manipulation was successful (as also evident from a greater rating by the repetition group of the two control questions, at $d = 1.56$ and $d = 0.82$ respectively). In line with expectations, the repetition group also provided a greater amount of unique details ($M_{diff} = $

4.82, $d = 0.52$) although results were not statistically significant ($p = .052$). Moreover, supplemental analyses show that the repetition group provided a greater amount of *total statements* ($M_{diff} = 14.17$, $d = 0.68$, see S1 File). This corroborates established findings that repetition increases retrieval (e.g. [11, 37]), and indicates that our manipulation of repetition (i.e. asking the participants to write down all they remembered about the witnessed event every second/third day during the two-week retention interval) was successful, albeit perhaps a bit on the weaker side.

## Limitations

A somewhat puzzling result in this study was the greater amount of effort cues for the repetition group at T1 compared to the no-repetition group. This is surprising because the experimental manipulation took place *after* the T1-session, and participants were randomized to each group. Thus, we would have expected similar levels of retrieval-effort cues for both groups at T1. This observed difference potentially indicates that there is quite a bit of variation *between* participants in terms of how many effort cues they use (as we have argued earlier–see Discussion in [9]). Therefore, it would be fruitful for future studies to look into individual differences in memory retrieval effort.

Regarding ecological validity, this experiment was conducted in the comfort (?) of a laboratory setting, with participants informed about the purpose of the study, and they could fully focus their attention on the staged crime video that they saw. This certainly contrasts with many real-life eyewitness experiences, where one might not be prepared for the witnessed event, may not experience great viewing conditions, and may have a looming threat to one's safety. Moreover, our participants were not explicitly exposed to influences from co-witnesses, post-event information or direct forms of misinformation. Our experiment does therefore not directly generalize to natural eyewitnesses. Nonetheless, we have investigated core processes of memory that do occur outside labs, that is, retention of episodic memories with and without repetition. Moreover, although our manipulation of repetition is not fully representative of spontaneous repeated reflections, we believe that it still effectively approximates the same outcome, namely increased memory strength. We therefore believe and hope that these results will be informative both for cognitive scientists and legal practitioners alike, even with these ecological limitations.

## Conclusion

In this experiment we set out to examine the relationship between retrieval effort and accuracy (and confidence), with a special focus on effects of time and repetition. Our results indicate that the retrieval-effort cues *Hedges* and *Delays* predict accuracy both directly after witnessing an event, and two weeks later, as they were continuously more common in incorrect responses. Confidence also predicted accuracy (higher for correct responses), even though repetition led to increased confidence for incorrect responses over time. Moreover, we found support for the idea that confidence is based on retrieval-effort cues, but results suggest additional factors beyond hedges and delays.

## Supporting information

**S1 Fig. Effects of accuracy, time and repetition on retrieval-effort cues in free and cued recall.**
(PDF)

**S2 Fig. Effects of accuracy, time and repetition on confidence and the retrieval-effort index in free and cued recall.**
(PDF)

**S1 File. Analyses of total amount of statements in testimonies.**
(PDF)

**S1 Table. Percentage distribution of ratings for each level of confidence across accuracy, time, and repetition.**
(PDF)

**S2 Table. Mediations between confidence and accuracy for each effort cue.**
(PDF)

## Author Contributions

**Conceptualization:** Philip U. Gustafsson.

**Formal analysis:** Philip U. Gustafsson.

**Funding acquisition:** Philip U. Gustafsson, Torun Lindholm.

**Investigation:** Philip U. Gustafsson.

**Methodology:** Philip U. Gustafsson, Torun Lindholm, Fredrik U. Jönsson.

**Project administration:** Philip U. Gustafsson.

**Resources:** Philip U. Gustafsson, Torun Lindholm.

**Supervision:** Torun Lindholm, Fredrik U. Jönsson.

**Writing – original draft:** Philip U. Gustafsson, Fredrik U. Jönsson.

**Writing – review & editing:** Torun Lindholm.

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
