## [Decision Letter · Decision Letter 0]

1 Mar 2022

PONE-D-22-00932Eyewitness Accuracy and Retrieval Effort: Effects of Time and RepetitionPLOS ONE

Dear Dr. Gustafsson,

Thank you for submitting your manuscript to PLOS ONE. After careful consideration, we feel that it has merit but does not fully meet PLOS ONE’s publication criteria as it currently stands. Therefore, we invite you to submit a revised version of the manuscript that addresses the points raised during the review process.

The manuscript has been assessed by two independent experts. As you can see from their comments, they see value in publishing this study, however they raise a number of important points that need to be addressed in a major revision. One major point raised by both reviewers is the lack of an effect of the main manipultations (i.e., repetition and time) on accuracy and detail reported. I agree that this is unusual, and would urge the authors to be as explicit and transparent as possible with respect to these failed manipulations, and to discuss the implications for interpreting the results. The reviewers also make important suggestions regarding the analyses, methods reporting and interpretation of the data which should be taken on board in a revision.

We look forward to receiving your revised manuscript.

Kind regards,

Maria Wimber

Academic Editor

PLOS ONE

Journal Requirements

(Compensation for participation in the study was supported by a grant from the Elisabeth and Herman Rhodin Memorial Foundation awarded to PUG. Compensation for the coders was supported by a grant from the Magnus Bergvall Foundation (www.magnbergvallsstiftelse.nu) awarded to TLÖ.)

Reviewers' comments:

Reviewer's Responses to Questions

**Comments to the Author**

1. Is the manuscript technically sound, and do the data support the conclusions?

Reviewer #1: Partly

Reviewer #2: Partly

2. Has the statistical analysis been performed appropriately and rigorously? 

Reviewer #1: Yes

Reviewer #2: No

3. Have the authors made all data underlying the findings in their manuscript fully available?

Reviewer #1: Yes

Reviewer #2: Yes

4. Is the manuscript presented in an intelligible fashion and written in standard English?

Reviewer #1: Yes

Reviewer #2: Yes

5. Review Comments to the Author

Reviewer #1: Previous research suggests that markers of retrieval effort can predict the accuracy of recalled information, with greater effort associated with reduced accuracy. This manuscript reports a single experiment intended to extend previous work on this topic and test how this relationship – and the relationships between retrieval effort, confidence, and accuracy – are moderated by factors likely to strength memory (i.e., repeated retrieval) and weaken memory (i.e., increased retention interval). These are manipulations of both applied and theoretical relevance. Fifty-six participants watched a simulated crime video and then completed free- and cued-recall questions relating to the content of the video. Before returning for a second recall session two weeks later, half of the participants engaged in four additional retrieve attempts.

Unfortunately, the manipulations of repetition and retention interval had no appreciable effect on the quantity or accuracy details recalled (i.e., effects were non-significant and negligible in size). Thus, it is not clear that memory was strengthened or weakened by these manipulations and, consequently, it is not clear that this work can extend previous work in the directions intended. However, this work does serve to replicate the core findings in previous work (i.e., demonstrating that markers of retrieval effort are negatively associated with the accuracy of retrieved information). And replication, in and of itself, is important.

However, before I could recommend publication, there are areas of the manuscript that require attention. I’ve listed these below. I hope that they will be useful to the authors if they choose to revise the manuscript.

1. Coverage of the relevant literature.

There are some relevant, and important, aspects of the literature that are omitted from the introduction. When considering factors associated with recall memory performance, it is really important to consider how metacognitive monitoring and control processes support the retrieval and reporting of information. We know that participant witnesses will strategically regulate report option (i.e., the decision to report or withhold retrieved information) and grain size/precision (i.e., the amount of detail reported) to balance competing demands for informativeness (provide as much information as possible) and accuracy (make the information you provide as reliable as possible). These metacognitive control processes affect the amount and accuracy of information volunteered (e.g., Fisher, 1996; McCallum et al., 2016), and confidence in the accuracy of volunteered information (with different confidence-accuracy relationships evident for fine- and coarse-grained details; e.g., Sauer & Hope, 2016; Weber & Brewer, 2008). Even if the researchers don’t want to include this as a design or analytical element of the current work, some consideration of how things like retrieval effort relate to (and may feed into) these underlying monitoring and control processes seems relevant.

Based on this literature, there are three implications for the current work (and, I think, for any work wishing to thoroughly understand the effects of manipulations on, and reliable indices of, recall accuracy):

First, ideally, the authors would code responses to provide details on how their manipulations affected the proportion of fine- and coarse-grained details provided. Second, ideally, markers of recall accuracy would be assessed differently for fine- and coarse-grained details. Finally, at a minimum, it is important to know (a) what instructions participants were given relating to the regulation of report option and grain size [were participants instructed, implicitly or explicitly, to prioritise accuracy or informativeness], and (b) how were answers coded for accuracy? For example, if the target wore a black top, would a response saying the top was “dark” be coded as correct or incorrect? Either approach is defensible – and there is certainly variability on how this issue has been treated in the past – so it’s important that reader (and those who may wish to replicate the work later) are clear on these details.

Further, in general, when motivating the research and considering the implications of findings, the authors need to be clearer in distinguishing between confidence and accuracy for recognition memory tasks and confidence and accuracy for recall memory tasks. There are similarities in the theoretical mechanisms at play, but there are also important differences. For example, inferential cues are prominent in theoretical explanations for confidence in recall (and associated metacognitive domains), but theories used to describe confidence processing for recognition tasks draw more heavily on signal detection theory and evidence accumulation models; emphasising the strength of the evidence underlying the decision rather than participants’ perceptions of the decision process.

2. Analysis

I really like the use of the multi-level modelling approach, and applaud the authors on using it. However, I think it will be unfamiliar to many readers of the eyewitness recall literature where repeated measures ANOVAs are still the “norm”. Thus, when presenting the results in Table 3, it may help readers appreciate the meaning of these findings if the authors provide some concrete guidance on interpreting the coefficients in the table. For example, for someone not familiar with interpreting logistic regression (or multi-level modelling), what do the data tell us about how the likely accuracy of a detail varies according to changes in the “hedges” variable (other than the obvious interpretation that the relationship is negative).

On a similar note, the results section contains a lot of output, for some reasonably complex models. I think it would help readers process this output, and appreciate the implications of the reported results, if the authors yoked the presentation of results to specific hypotheses/research questions. “We hypothesised, X. If this hypothesis were borne out in the data, we would expect to see…” then you can present the outputs and it becomes very easy for the reader to appreciate how the reported outputs align with expectations. I think steps like these become increasingly important when presenting unfamiliar analyses

Also: Was participant included as a random factor in these models? Did the models allow for both random intercepts and random slopes, or only for random intercepts?

I’m not sure it’s appropriate to group free and cued recall outputs together for analyses. These are different types of memory task, likely to rely differently on the underlying metacognitive monitoring and control processes, and requiring (and potentially facilitating) different levels of effort by the retriever. Collapsing these data may be fine, but some justification for this analytical choice is required.

3. Discussion and interpretation

Although I don’t necessary disagree with the authors’ interpretations of their findings, I was struck by the absence of early, explicit consideration of the failed manipulations. The authors need to be careful not to over-interpret their null effects (e.g., of repetition and retention interval on memory performance) and, instead, think about how the absence of these predicted effects might place important caveats on the other conclusions they draw (e.g., relating to effects on effort, confidence, and accuracy). It is a little odd that neither of these typically robust mechanisms showed any effects on recall performance here. Why might that be / what might that indicate? Again, I’d recommend reading Fisher’s (1996) paper in Behavioural and Brain Sciences. It’s a great example of how a seemingly failed manipulation might indicate something of interest in the data. Further, I was interested by the fact that, in the limitations section, there was no mention of the fact that the two key manipulations failed to produce any appreciable effect on memory performance (i.e., the quantity or accuracy of recalled information). This seems like something worthy of comment. In the introduction, the authors set up the paper as an investigation of how time and repetition may affect the relationship between accuracy and memory retrieval effort based on the weaking (via increased retention interval) or strengthening (via repetition) of memories. Moreover, as the authors note in their conclusion: They set out with a special focus on the effects of time and repetition. Thus, their unsuccessful manipulations of both time and repetition would seem to constrain their ability to address the key foci of the experiment. This is not to say that other aspects of the data can’t provide a valuable contribution to the literature; but I’d expect some consideration of these failed manipulations, and how they might colour interpretations of the observed effects.

Minor points

* I wondered why “hedges” were considered an index of effort. They seem, intuitively, to be markers of uncertainty (and therefore conceptually related to confidence). Can the authors elaborate on why these are best viewed as cues associated with effort rather than cues indicating uncertainty?

* The claim that confidence judgements seem to be especially unreliable when made a considerable time after the event (line 120) needs some support (e.g., citations). Typically, the predictive value of confidence varies according to the interval between the decision/retrieval effort and confidence, rather than the interval between encoding and confidence (see Palmer et al., 2013; Sauer et al., 2010; Wixted et al., 2016, for examples). It’s true that overconfidence can increase with retention interval, but that’s not to say that confidence is unreliable. Perhaps this also depends on whether one looks at the recall or recognition literatures?

* On line 469, the authors conclude that “retrieval effort decreased over time…”. Given that accuracy did not, this poses a conundrum: How were participants able to maintain accuracy, despite decreased effort, as retention interval increased? I think the answer probably relates to the operationalisation of retrieval effort in this claim. Specifically, I’m not sure that we can equate “retrieval efforts cues” to retrieval effort. Thus, although the presence of retrieval effort cues might decrease over time, I’m not sure it can be concluded that retrieval effort decreased. Cues are just that: Cues. They may be indicative of underlying effort, but they are not direct indexes of the underlying construct. This is, in some sense, a wording issue but it is conceptually important.

* On a related note: If accuracy doesn’t change over time, but indices of retrieval effort decreases over time, it seems there might be some need to qualify (if only slightly) the conclusion that retrieval effort predicts accuracy over time (i.e., the two must, to some extent, start to dissociate).

Reviewer #2: Review for PLOS One

Eyewitness accuracy and retrieval effort: Effects of time and repetition

This manuscript presents one experiment testing whether retrieval-effort cues are inflated among eyewitnesses both immediately after a witnessed event and after a two-week delay. Half of the witnesses were induced to repeat their recollections via an online survey; the other half were not. Both groups were interviewed immediately after the event and after a two-week delay. The manuscript is well-written and provides useful data on how repetition and delay implicate memory accuracy as a function of retrieval effort cues.

The authors should include an important unmentioned explanation for why confidence increases over time: post-identification feedback (e.g., Wells & Bradfield, 1998). See pp. 6-7

The authors should also consider including reference to retrieval-enhanced suggestibility (e.g., Chan, Wilford, & Hughes, 2012).

One potential issue with these data is that both groups showed decreased retrieval effort at Time 2, even though the manipulation check items indicated that the repetition group did rehearse their memories more so than the control group. The authors attribute this null finding to context-dependent learning in which the control group’s recall was boosted by being in the same environment at Time 1 and Time 2. This hypothesis bears pursuing because the repetition group also benefitted (presumably) from context-dependent learning. Is it the authors’ contention that context-dependent learning retrieval effort produced a floor effect of retrieval effort such that repetition couldn’t have any additional effect? I wonder if these data would be more useful with the addition of another experiment where the context-dependent learning explanation can be eliminated.

I was also surprised that there was no effect of time on number of accurate statements or total number of accurate details, nor was there any interaction between repetition and time (page 14). This seems counterintuitive to previous research showing that memory decays over time. Using this finding to argue that “confidence can remain a stable predictor over time” (page 24) is unwise because this seems to be a particularly unique scenario in which there was no memory loss over two weeks.

One other issue is that the authors dropped non-word fillers from predictive analyses examining the effect of time and repetition on the relationship between accuracy and retrieval effort. Given that this DV was significant (albeit in the opposite direction than predicted), it seems like some additional analyses are warranted. Could it not have been included in the model as a reciprocal predictor, as in, fewer non-word fillers predicting accuracy? Couldn’t this significant predictor explain some of the “leftover variance” (page 24)?

I think the authors need to be more cautious in their conclusions about believing confident witnesses (page 23). Their argument is only appropriate if intervening influences are not present (e.g., post-identification feedback, influence from co-witnesses).

I would like the authors to address how to use these data in the field. Namely, how could information about hedges and delays help individual jurors or investigators evaluate particular witnesses? Is there some criterion of hedge/delay behavior beyond which witnesses become untrustworthy? Probably not, but it would be useful to have the authors speculate on what an individual evaluator should do with this information.

Minor point:

Page 12 – there are kappa values for the second and third exact overlap percentages, but not the first (line 268)

6. PLOS authors have the option to publish the peer review history of their article (what does this mean?). If published, this will include your full peer review and any attached files.

Reviewer #1: No

Reviewer #2: No

---

## [Author Response · Author response to Decision Letter 0]

10 Jun 2022

We thank the reviewers for the helpful comments. We hope that all the corrections and revisions will be satisfactory. Please see the attached file "Response to reviewers" for the responses.

Kindly,

Philip

---

## [Decision Letter · Decision Letter 1]

9 Aug 2022

Eyewitness Accuracy and Retrieval Effort: Effects of Time and Repetition

PONE-D-22-00932R1

Dear Dr. Gustafsson,

We’re pleased to inform you that your manuscript has been judged scientifically suitable for publication and will be formally accepted for publication once it meets all outstanding technical requirements.

Kind regards,

Jonathan Jong, PhD

Academic Editor

PLOS ONE

Additional Editor Comments (optional):

Reviewers' comments:

Reviewer's Responses to Questions

**Comments to the Author**

1. If the authors have adequately addressed your comments raised in a previous round of review and you feel that this manuscript is now acceptable for publication, you may indicate that here to bypass the “Comments to the Author” section, enter your conflict of interest statement in the “Confidential to Editor” section, and submit your "Accept" recommendation.

Reviewer #1: All comments have been addressed

2. Is the manuscript technically sound, and do the data support the conclusions?

Reviewer #1: Yes

3. Has the statistical analysis been performed appropriately and rigorously? 

Reviewer #1: Yes

4. Have the authors made all data underlying the findings in their manuscript fully available?

Reviewer #1: Yes

5. Is the manuscript presented in an intelligible fashion and written in standard English?

Reviewer #1: Yes

6. Review Comments to the Author

Reviewer #1: I apologise the delay in returning my review. I commend the authors of their conscientious approach to addressing the previous round of reviews. I have no remaining concerns.

Although I think it would be ideal to re-examine the data (if the raw data are available) to see if a shift in gran-size accounts for the absence of effects on accuracy, I would not demand this of the authors.

Similarly, although I wouldn't combine free recall data with cued recall data for analysis (and there do appear to be some differences in the data), this is perhaps also a matter of personal preference.

7. PLOS authors have the option to publish the peer review history of their article (what does this mean?). If published, this will include your full peer review and any attached files.

Reviewer #1: No
